Collaborative networks enable the rapid establishment of serological assays for SARS-CoV-2 during nationwide lockdown in New Zealand

McGregor Reuben 1 2
Whitcombe Alana L. 1 2
Sheen Campbell R. 3
Dickson James M. 4
Day Catherine L. 5
Carlton Lauren H. 1
http://orcid.org/0000-0002-6932-8472 Sharma Prachi 1
http://orcid.org/0000-0003-3660-452X Lott J. Shaun 2 4
Koch Barbara 3
Bennett Julie 6
http://orcid.org/0000-0002-1865-1536 Baker Michael G. 6
Ritchie Stephen R. 1 7
Fox-Lewis Shivani 8
Morpeth Susan C. 9
Taylor Susan L. 9
Roberts Sally A. 2 8
Webb Rachel H. 1 10
http://orcid.org/0000-0001-6548-637X Moreland Nicole J. 1 2 n.moreland@auckland.ac.nz
1 Faculty of Medical and Health Sciences, University of Auckland , Auckland , New Zealand
2 Maurice Wilkins Centre, University of Auckland , Auckland , New Zealand
3 Protein Science and Engineering, Callaghan Innovation , Christchurch , New Zealand
4 School of Biological Sciences, University of Auckland , Auckland , New Zealand
5 Department of Biochemistry, University of Otago , Dunedin , New Zealand
6 Department of Public Health, University of Otago , Wellington , New Zealand
7 Infectious Diseases Department, Auckland City Hospital , Auckland , New Zealand
8 Department of Microbiology, LabPLUS, Auckland City Hospital , Auckland , New Zealand
9 Middlemore Hospital , Auckland , New Zealand
10 Starship Children’s Hospital , Auckland , New Zealand
Krajaejun Theerapong
Electronic publication date: 2020 Sep 3
Publication date: 2020
Volume: 8
Electronic Location ID: e9863
Received 2020 Jul 12; Accepted 2020 Aug 13
Copyright: © 2020 McGregor et al.
Copyright year: 2020
Copyright holder: McGregor et al.
License: This is an open access article distributed under the terms of the Creative Commons Attribution License, which permits unrestricted use, distribution, reproduction and adaptation in any medium and for any purpose provided that it is properly attributed. For attribution, the original author(s), title, publication source (PeerJ) and either DOI or URL of the article must be cited.
License URL: https://creativecommons.org/licenses/by/4.0/

Keywords: COVID-19, Serology, SARS-CoV-2, Neutralising antibodies, Spike protein

Funding: School of Medicine Foundation (University of Auckland) COVID-19 Innovation Acceleration Fund (MBIE) Callaghan Innovation Strategic Investment Fund This work was funded by the School of Medicine Foundation (University of Auckland), the COVID-19 Innovation Acceleration Fund (Ministry of Business, Innovation and Employment) and Callaghan Innovation Strategic Investment Fund. The funders had no role in study design, data collection and analysis, decision to publish, or preparation of the manuscript.

==============================
Background

Serological assays that detect antibodies to SARS-CoV-2 are critical for determining past infection and investigating immune responses in the COVID-19 pandemic. We established ELISA-based immunoassays using locally produced antigens when New Zealand went into a nationwide lockdown and the supply chain of diagnostic reagents was a widely held domestic concern. The relationship between serum antibody binding measured by ELISA and neutralising capacity was investigated using a surrogate viral neutralisation test (sVNT).

Methods

A pre-pandemic sera panel (n = 113), including respiratory infections with symptom overlap with COVID-19, was used to establish assay specificity. Sera from PCR‑confirmed SARS-CoV-2 patients (n = 21), and PCR-negative patients with respiratory symptoms suggestive of COVID-19 (n = 82) that presented to the two largest hospitals in Auckland during the lockdown period were included. A two-step IgG ELISA based on the receptor binding domain (RBD) and spike protein was adapted to determine seropositivity, and neutralising antibodies that block the RBD/hACE‑2 interaction were quantified by sVNT.

Results

The calculated cut-off (>0.2) in the two-step ELISA maximised specificity by classifying all pre-pandemic samples as negative. Sera from all PCR-confirmed COVID-19 patients were classified as seropositive by ELISA ≥7 days after symptom onset. There was 100% concordance between the two-step ELISA and the sVNT with all 7+ day sera from PCR‑confirmed COVID-19 patients also classified as positive with respect to neutralising antibodies. Of the symptomatic PCR-negative cohort, one individual with notable travel history was classified as positive by two-step ELISA and sVNT, demonstrating the value of serology in detecting prior infection.

Conclusions

These serological assays were established and assessed at a time when human activity was severely restricted in New Zealand. This was achieved by generous sharing of reagents and technical expertise by the international scientific community, and highly collaborative efforts of scientists and clinicians across the country. The assays have immediate utility in supporting clinical diagnostics, understanding transmission in high-risk cohorts and underpinning longer‑term ‘exit’ strategies based on effective vaccines and therapeutics.

Introduction

Severe acute respiratory syndrome coronavirus 2 (SARS-CoV-2), the causative agent of the Coronavirus Disease 2019 (COVID-19) global pandemic, is typically detected in acutely infected individuals via nucleic acid-based polymerase chain reaction (PCR) tests (Sethuraman, Jeremiah & Ryo, 2020). The first evidence of community transmission in New Zealand was reported on 23 March 2020, the country went into an intense ‘Alert Level 4’ lockdown (the highest level of a 4-level response system) three days later and remained at this Alert Level for the following five weeks (Baker, Kvalsvig & Verrall, 2020). During this time there was notable increase in national nucleic acid testing capacity and this remains the cornerstone of SARS-CoV-2 diagnosis. However, there is also a need for reliable serological assays that measure antibody responses to the virus. While serological assays are not suited to the diagnosis of acute infections due to the days required to generate an antibody response, they are critical for determining past exposure and investigating immune responses (Abbasi, 2020; Sethuraman, Jeremiah & Ryo, 2020; Krammer & Simon, 2020).

Numerous laboratory-based serological assays for SARS-CoV-2 are being developed worldwide including enzyme-linked immunosorbent assays (ELISAs) and bead-based immunoassays for measurement of SARS-CoV-2 antibodies (Petherick, 2020; Garcia-Basteiro et al., 2020; Krammer & Simon, 2020), as well as virus neutralisation assays for quantification of neutralising antibodies (Tan et al., 2020; Anderson et al., 2020). The spike protein (S protein) expressed by SARS-CoV-2 contains a receptor binding domain (RBD), which interacts with host cells via the human angiotensin-converting enzyme 2 (hACE2) (Diamond & Pierson, 2020). The S protein is highly immunogenic and, given the integral role of the S protein and RBD in facilitating viral entry, these antigens form the basis of many immunoassays described to date (Duan et al., 2020; Garcia-Basteiro et al., 2020; Long et al., 2020; Amanat et al., 2020). Recent data from immunoassays based on the SARS-CoV-2 nucleocapsid protein also show high sensitivity (Sethuraman, Jeremiah & Ryo, 2020; Bryan et al., 2020). However, the higher sequence conservation of the SARS-CoV-2 nucleocapsid with other coronaviruses, compared to the S protein, increases the possibility of antibody crossreactivity against the nucleoprotein in those infected with related viruses (Krammer & Simon, 2020; Anderson et al., 2020).

As New Zealand entered Alert Level 4 Lockdown, and the supply chain of diagnostic reagents for managing COVID-19 testing was a widely held domestic concern, we sought to establish a serologic ELISA assay based on locally-produced SARS-CoV-2 antigens. We adapted the two-step ELISA protocols developed at The Icahn School of Medicine at Mount Sinai (New York City) based on the S protein and RBD, which has FDA Emergency Use Authorization (Amanat et al., 2020; Stadlbauer et al., 2020). A panel of sera/plasma from PCR-confirmed COVID-19 patients was compared with prepandemic sera to determine assay parameters. The relationship between antibody binding to the S protein and RBD, and neutralising ability was explored using the surrogate viral neutralisation test (sVNT) recently developed at Duke-NUS (Singapore) (Tan et al., 2020). Finally, the utility of both the ELISA and sVNT to identify prior SARS-CoV-2 infection was investigated in a cohort of PCR-negative patients that presented to hospital with respiratory symptoms during the lockdown period.

Methods

Human samples

Human plasma and sera were obtained from several different sources, all of which were granted ethical approval by the University of Auckland Human Ethics Committee or the Health and Disability Ethics Committee in New Zealand. A total of 113 samples collected before 2020 were used as negative controls and all participants (or their parents or legal 109 guardians) provided written informed consent. These included healthy adult volunteers (n = 31) (ethics UOA021200), hospitalised adults with bacteraemia or bacterial pneumonia (n = 25) (ethics HDEC 17/STH/233), together with children infected with various respiratory viruses (n = 57) (ethics HDEC 17/NTA/262) (Bennett et al., 2019). The patients with bacterial pneumonia had signs, symptoms and radiological imaging diagnostic of pneumonia and Streptococcus pneumoniae identified. The COVID-19 panel comprised serum and plasma (n = 21) obtained from 17 patients with PCR-confirmed SARS-CoV-2 infection and those with respiratory symptoms fitting the case definition for COVID-19 testing that were subsequently found negative by PCR (n = 82). The PCR-confirmed and PCR-negative patients were admitted to Middlemore or Auckland City hospitals in Auckland, New Zealand between March and May 2020 with residual sera/plasma stored following completion of all routine testing for validation of antibody diagnostics (ethics HDEC 20NTB76) (Table 1). All serum and plasma samples were heated at 56 °C for 30 min before use to inactivate any residual virus, as published (Amanat et al., 2020). Pooled human intravenous immunoglobulin (IVIG) were produced before 2020 from >1,500 donors in New Zealand (Intragram P) and Europe and North America (Privigen) (CSL Behring).

Table 1 Patient demographics.

The pre-pandemic panel comprised paediatric respiratory infections, adults hospitalised with bacteraemia or bacterial pneumonia and healthy adult laboratory donors. The pandemic samples comprised PCR-confirmed COVID-19 and symptomatic PCR-negative groups.

		Healthy donors	Respiratory infections	Hospitalised infections	COVID-19 cases	Symptomatic PCR-negative	
Total		31	57	25	17	82	
Age	Median	>20 years	10	67	48	60	
	Range		5–14	19–93	23–86	17–94	
Gender	M/F	n/a*	30/27	14/11	7/10	43/39	
Year collected		2014–2019	2018–2019	2018	2020	2020	
Viral infections	RSV		6				
	Influenza A		14				
	Influenza B		16				
	Parainfluenza 1		1				
	Rhino/enterovirus		7				
Other infections	Bacterial pharyngitis		2				
	Pertussis		4				
	Mycoplasma		7				
	Bacterial pneumonia			5			
	Bacteraemia			20			
Note:

* Not available.

Indirect ELISA

The S protein and RBD antigens were expressed and purified from pCAGGS-RBD and pCAGGS-solSpike vectors respectively, kindly provided by Florian Krammer (The Icahn School of Medicine at Mt Sinai, New York City, NY, USA) using Expi293F or Freestyle293 human embryonic kidney (HEK) cells and published protocols (Stadlbauer et al., 2020), but with a modified transient transfection protocol using polyethyleneimine (PEI). Plasmid DNA was added at 3.5 μg/mL with PEI 7.0 at μg/mL for 24 h, after which culture volumes were doubled and supplemented with 2.2 mM valproic acid. Cultures were incubated with shaking for a further 72 h before protein purification was performed.

The two-step ELISA protocol that includes a single point screen against the RBD, followed by a confirmatory titration against the S protein (Stadlbauer et al., 2020), was utilised with minor modifications. In step one immunoplates were coated with RBD (5 μg/ml) overnight at 4 °C and blocked with phosphate-buffered saline supplemented with 0.1% Tween 20 (PBST) and 3% skim milk powder at 20 °C for 1 h. Serum or plasma diluted 1:100 in diluent buffer (PBST + 1% skim milk powder) was added for 1 h at 20 °C. Following washing (3× PBST), peroxidase-labelled anti-human IgG (97221; Abcam, Cambridge, United Kingdom) diluted 1:10,000 was added for 1 h at 20 °C. The reaction was developed with 3,3′,5,5′-Tetramethylbenzidine (TMB) and stopped with 1M HCl. The optical density (OD) at 450–570 nm was measured using an EnSight absorbance reader. In step two, immunoplates were coated with S protein (5 μg/ml) overnight at 4 °C and the ELISA was performed using the same protocol as in step one, except that 3-fold serial dilutions of samples starting at 1:100, were prepared. Samples were classified as seropositive if they had an OD above the calculated cutoff (>0.2) in the single point RBD ELISA and in at least two consecutive wells in the S protein titration ELISA. Positive and negative quality controls were included on each plate, with the assay meeting acceptance criteria if the OD was >0.75 and <0.03 for the positive and negative control, respectively.

To assess healthy control IgG reactivity to human coronaviruses (HCoV) ELISA were performed as described above with S1 antigens from HKU1, NL63, 229E and SARS-CoV-2 (Sino Biological, Beijing, China) coated at 5 μg/ml and a 1:300 sera dilution. Samples from PCR-confirmed COVID-19 patients were also subject to isotype-specific titration ELISA using peroxidase-labelled anti-human IgG (97221; Abcam, Cambridge, United Kingdom), anti-human IgM (97205; Abcam, Cambridge, United Kingdom) and anti-human IgA (97215; Abcam, Cambridge, United Kingdom) at 1:10,000 dilution. The area under the curve (AUC) was used to compare isotype-specific antibody titres and was calculated by subtracting background AUC of pooled negative control sera for each isotype in Prism 8 (GraphPad).

Surrogate viral neutralisation test

Surrogate neutralisation assays were carried out using a SARS-CoV-2 sVNT Kit supplied pre-launch by GenScript as described (Tan et al., 2020). Briefly, serum or plasma was diluted 1:20 before incubation with an equivalent volume of peroxidase-conjugated RBD for 30 min at 37 °C. This was added to wells pre-coated with human ACE-2 receptor protein and incubated for a further 15 min at 37 °C. Following washing and TMB development the OD 450 nm was measured using an EnSight absorbance reader. Inhibition was calculated as (1—OD sample/OD of negative control) × 100. Samples with a percentage inhibition ≥20% were deemed to have neutralising antibodies.

Statistical analysis

The two-step ELISA and sVNT analysis utilised a Kruskal–Wallis test for comparison between three groups with Dunn’s multiple comparisons test. AUC data were log10 transformed to achieve a Gaussian distribution and analysed by one-way ANOVA followed by Tukey’s multiple comparisons test. Correlations were calculated using Pearson’s correlation coefficient. Data were analysed using Prism 8 (GraphPad) or R (version 3.6.3) within R Studio (version 1.2.5033) and a P value of ≤0.05 was considered statistically significant.

Results

Indirect ELISA with the RBD and S protein

The RBD and S proteins used as antigens in the ELISA were shown to be >95% pure by SDS-PAGE following expression and purification from mammalian (HEK derived) cells. In line with previous reports the yield of RBD was approximately 10-fold higher per litre of culture than the S protein (Amanat et al., 2020). To establish ELISA cut-off values a panel of 113 sera collected prior to 2020 were tested, with the cut-off defined as mean plus three-standard deviations. Importantly, this panel included samples from participants with bacterial pneumonia and common respiratory viruses that have symptom overlap with SARS-CoV-2 infections (Table 1). The healthy control sera (n = 31) within the panel showed broad reactivity with S protein antigens from HCoV (HKU1, NL63, 229E), but not for SARS-CoV-2 (Fig. S1). As shown in Fig. 1A, all serum samples from PCR confirmed COVID-19 patients collected ≥7 days from symptom onset had IgG above the determined cut-off in the RBD screening ELISA (P < 0.0001), while the pooled IVIG preparations (representing >1,500 donors each) were negative.

Figure 1 Antibody responses in pre-pandemic controls and PCR confirmed COVID-19 sera.

(A) Screening IgG ELISA against RBD for pre-pandemic controls (orange), IVIG (pink squares) and PCR confirmed COVID-19 sera <7 (light blue) and 7+ days from symptom onset (royal blue). Samples boxed in grey above the cut-off (red dashed line) were titrated against the spike protein. (B) Example of the confirmatory IgG titration ELISA against Spike protein. Samples above the cut-off (dashed line) in at least two consecutive dilutions are deemed seropositive (royal blue). Isotype specific ELISA against the RBD (C) and Spike (D) for IgG (blue), IgM (green) and IgA (red) in PCR confirmed COVID-19 sera <7 and 7+ days from symptom onset. One patient showed higher IgM than IgG responses and is indicated (#). AUC, area under the curve. Illustrations created with BioRender.com.

To illustrate the utility of the second confirmatory ELISA, samples with absorbance close to the cut-off in the RBD screen were titrated against the S protein. As shown in Fig. 1B, the anti-S protein IgG titration clearly separated the true positives from the false positives, as only the PCR-confirmed COVID-19 samples met the seropositive criteria (OD > cut-off in at least two consecutive dilutions). Indeed, this second ELISA resulted in all pre-pandemic samples being classified as negative and a calculated specificity of 100%, albeit in a moderately sized sample panel. The use of IgM and IgA in the two-step ELISA protocol were also explored, however IgM was found to have lower sensitivity with only 4/18 of the 7+ day COVID-19 samples being seropositive, compared with 18/18 (100%) for IgG. IgA was deemed unsuitable as the assay window was inferior to that of IgG (OD range 0.01–0.82 compared with 0.01–1.42), and recent reports highlighted reduced sensitivity and specificity for IgA based SARS-CoV-2 ELISA (Meyer et al., 2020; Beavis et al., 2020).

Isotyping ELISA performed with the PCR-confirmed COVID-19 samples showed significantly higher titres for IgG compared with IgM and IgA for both the RBD and S protein (Figs. 1C and 1D, P < 0.01). Of note, only one sample had IgM levels higher than IgG, with the remaining 20 samples having higher IgG than IgM, despite ~40% of the samples being collected within 2-weeks of symptom onset.

Neutralising anti-SARS-CoV-2 antibodies

The presence of neutralising antibodies (NAbs) capable of blocking the interaction between the SARS-CoV-2 RBD and the hACE-2 receptor was assessed using sVNT (Fig. 2A) (Tan et al., 2020). To validate performance, the pre-pandemic panel (n = 113) was compared with the PCR-confirmed COVID-19 samples. Using the cut-off value of 20% inhibition recommended by the manufacturer, all control samples tested were negative, resulting in a 100% specificity. Similarly, all sera from PCR-confirmed COVID-19 patients collected ≥7 days from symptom onset were positive for a sensitivity of 100%. An experimentally calculated cutoff of 19.59% (mean + 3SD of controls) also gave 100% sensitivity and there was a significant increase in the level of NAbs (% inhibition) in the 7+ day COVID-19 samples compared with controls (P < 0.001).

Figure 2 Surrogate viral neutralisation test (sVNT).

(A) Negative controls (orange) and PCR confirmed COVID-19 sera <7 (light blue) and 7+ days from symptom onset (royal blue). Samples with inhibition above the cut-off (red dashed line, 20%) were deemed positive. Following validation, the sVNT was run on PCR negative (ND) samples with COVID-like symptoms (grey). One sample showed 65.5% inhibition (green) indicating positive SARS‑CoV-2 neutralisation. (B) Pearson correlation for % inhibition (sVNT) and IgG antibody titre to RBD in PCR confirmed COVID-19 sera (n = 21). Inset, Pearson correlation coefficients comparing % inhibition (sVNT) and antibody isotype (IgG, IgA and IgM responses) to RBD (left) and spike protein (right). Colour scale from weak correlation (Pearsons coefficient of 0, white) to strong correlation (Pearson’s coefficient of 1, red). (C) % inhibition (sVNT) for temporal samples from three patients with PCR confirmed COVID-19 infection. (D) Pearson correlation for % inhibition (sVNT) and days since symptom onset for PCR confirmed COVID-19 sera (n = 21). AUC, area under the curve; ND, not detected.

A correlation analysis of the sVNT with ELISA titre data for IgG, IgM and IgA against the RBD and S protein found the highest correlation between the sVNT and anti-RBD IgG (r = 0.91, P < 0.0001), suggesting anti-RBD IgG is the best predictor of neutralisation (Fig. 2B). Temporal samples were available for three of the PCR-confirmed COVID-19 patients, with increasing NAbs detected in patient one between days 6 and 9, and higher levels of NAbs in patients two and three between days 11 and 31 and days 15 and 31, respectively (Fig. 2C). In keeping, there was a significant, positive correlation between days from symptom onset and the level of NAbs (% inhibition) in the PCR-confirmed COVID-19 patient group up to 40 days (Fig. 2D; r = 0.54, P < 0.05).

Patients with COVID-19-like symptoms

To assess the utility of the two-step ELISA protocol and the sVNT in detecting prior SARS-CoV-2 infection, 82 patients who presented to hospital during the lockdown period with respiratory symptoms fitting the case definition for COVID-19 testing, but negative for PCR, were analysed. A single patient was classified as seropositive in the two-step ELISA (RBD screen, OD 0.87; S protein titration OD > 0.2 in four consecutive dilution wells, Supplemental Data). Similarly, only one patient was classified as positive in the sVNT with 65.5% inhibition (Fig. 2A). This was the same patient identified using the two-step ELISA, indicating a prior undetected SARS-CoV-2 infection. Although the patient was PCR negative at presentation, they had notable travel history as a risk factor.

Discussion

At the time of writing, New Zealand has eliminated community transmission of SARS-CoV-2. This is a very different scenario to that during Alert Level 4 Lockdown, when human activity was severely restricted, and the perceived need for rapid establishment of serological assays was significant. Through the generous sharing of reagents and technical expertise by the international scientific community, and highly collaborative efforts of scientists and clinicians across the country we were able to establish and assess the described serological assays in a matter of weeks. These assays detect SARS-CoV-2 IgG and the presence of neutralising antibodies, in persons who have been infected with SARS-CoV-2 at least seven days after symptom onset, with repeat sampling recommended in those where samples are obtained <7 days from symptom onset. While the lack of community transmission in New Zealand has limited the scale of serological investigations thus far, maintaining our current state requires near-perfect management of returning travellers, as new cases are imported. Serological assays could have a key role in our border setting, identifying persons who have previously had a SARS-CoV-2 infection. Furthermore, the application of serological assays that enable accurate measurement and further understanding of SARS-CoV-2 immune responses are crucial to any ‘exit strategy’ reliant on effective vaccines and/or therapeutics (Baker, Kvalsvig & Verrall, 2020).

The two-step ELISA protocol is based on published protocols (Stadlbauer et al., 2020), with the seropositive cut-off established using a panel of pre-pandemic sera that showed bacterial pneumonia and other common respiratory infections such as rhinovirus, influenza and respiratory syncytial virus did not cross-react with the SARS-CoV-2 RBD and S proteins. Although the pre-pandemic panel does not include samples from known human coronavirus infections, the healthy control sera in the panel had broad reactivity with HCoV antigens. This is consistent with other studies, which have reported that the majority of banked pre-pandemic sera react with human coronavirus antigens given the ubiquitous nature of these infections (Amanat et al., 2020; Juno et al., 2020). Furthermore, neither of the pooled IVIG preparations derived from >1,500 donors tested in this study reacted with SARS-CoV-2 RBD or S proteins, consistent with the negligible crossreactivity of human coronavirus sera reported by the assay developers (Amanat et al., 2020; Tan et al., 2020).

All PCR-confirmed COVID-19 patients showed strong seroconversion a week after symptom onset, with mean ELISA AUC titre for anti-RBD IgG of 1:1,500. Although our sample size was limited, the antibody responses follow trends observed in larger COVID-19 cohorts in settings with higher case numbers. This includes a concurrent rise in IgM and IgG (Huang et al., 2020; To et al., 2020), and robust antibody responses to the full-length S protein, as well as RBD (Chen et al., 2020; Amanat et al., 2020; Juno et al., 2020). The level of NAbs, which block the interaction of the RBD with the hACE-2 receptor was highly correlated with anti-RBD IgG in this study. This is in keeping with observations that most SARS-CoV-2 neutralising epitopes are localised in the RBD (Ju et al., 2020; Chen et al., 2020) and that neutralisation measured by conventional, viral neutralisation assays is reported to correlate with the sVNT (Tan et al., 2020). While more extensive comparison of the sVNT with gold standard viral neutralisation assays is required, the ability of the assay to measure total NAbs in less than 2 h, without the need to handle live SARS-CoV-2 virus, provides strong rationale to consider these types of assays in pandemic management, particularly in settings where BSL3 laboratory infrastructure is limited.

Serological tests are traditionally used to support clinical diagnosis by determining recent or prior infection when swab results are negative (Bryant et al., 2020). In the context of SARS-CoV-2 infections, serology could confirm diagnosis in individuals who are PCR negative due to late presentation or technical limitations of swab-based tests. Proof of principle was demonstrated in this study by the identification of a seropositive individual in the two-step ELISA and the sVNT from a cohort of 82 patients with respiratory symptoms that presented to hospital during the Alert level 4 lockdown period. The positive serology results, combined with travel history, suggest a prior undetected SARS-CoV-2 infection and highlight the value of accurate serology in clinical diagnosis. Extending the application of serology to understand SARS-CoV-2 transmission within clusters was recently demonstrated in Singapore, where detection of seropositive individuals enabled three small clusters to be connected (Yong et al., 2020). Beyond transmission studies, serological testing in managed isolation facilities could provide a more complete picture of previous SARS-CoV-2 infections in returning citizens in settings like New Zealand where strict border restrictions remain.

Large scale serosurveys require careful consideration of assay sensitivity and specificity. While the serological assays described in this study show very high sensitivity and specificity, markedly larger cohorts would need to be tested before robust assessments of accuracy can be performed. Indeed, the need for rigorously validated assays is arguably greater in low prevalence settings like New Zealand before wide-spread serosurveys are considered. Even an assay with near perfect specificity of 99.9% would identify 100 false positives in every 100,000 individuals, and with <0.1% of the New Zealand population having been infected, positive predictive value is extremely limited. In contrast, targeted studies of high-risk individuals such as health care workers, those linked with clusters and in managed isolation would provide an opportunity to assess serological assay performance, and generate critical local data on levels of antibodies in those with symptomatic versus asymptomatic infection (Krammer & Simon, 2020; Bryant et al., 2020). Studies that incorporate RBD-based assays such as those described here, as well as the nucleoprotein as per the high-throughput systems recently launched by Roche and Abbott (Bryan et al., 2020), would provide insight into SARS-CoV-2 exposure together with antibody neutralisation, and lay the foundation for future studies aimed at understanding long-term antibody persistence and vaccine efficacy.

Conclusion

In summary, the collective support of international colleagues combined with a collaborative domestic network of scientists and clinicians enabled serological assays for COVID-19 to be established during a nationwide lockdown. The success of the open approach we have taken may offer considerable advantages in other settings where access to reagents and resources is currently constrained. Importantly, these assays showed that hospitalised patients infected with SARS-CoV-2 develop high levels of neutralising antibodies. While the low prevalence of COVID-19 infections in New Zealand currently limits the use of serological assays in population level serosurveys, they have immediate utility in clinical diagnostics, studies to understand transmission in high-risk cohorts and underpinning longer-term ‘exit’ strategies based on effective vaccines and/or therapeutics.

Supplemental Information

Supplemental Information 1 Serological data obtained in the two-step ELISA and surrogate viral neutralisation tests.

The data for the RBD ELISA are shown as optical density (OD), and the data for the Spike ELISA are shown as titration. The sVNT data is percentage inhibition (%).

Click here for additional data file.

Supplemental Information 2 Serum IgG reactivity of pre-pandemic healthy adult controls with human coronavirus S proteins.

ELISA were performed with sera (n = 31) against S1 proteins of one β- (HKU1) and two α-HCoVs (NL63 and 229E) as well as S1 and Spike proteins of SARS-CoV2. The cutoff of OD >0.2 is indicated for reference to SARS-CoV2 spike protein ELISA. IVIG is indicated as pink squares.

Click here for additional data file.

We thank Shirley Lawrence and Franklin Han for assistance with patient recruitment, and Paul Austin for laboratory support. Florian Krammer is thanked for the timely provision of vectors for expression of the proteins used in this study. We are grateful to Linfa Wang at Duke-NUS and GenScript for providing sVNT testing kits and technical advice.

Additional Information and Declarations

Competing Interests

Author Contributions

Human Ethics

Data Availability

The authors declare that they have no competing interests.

Reuben McGregor conceived and designed the experiments, performed the experiments, analysed the data, prepared figures and/or tables, authored or reviewed drafts of the paper, and approved the final draft.

Alana L. Whitcombe conceived and designed the experiments, performed the experiments, analysed the data, prepared figures and/or tables, authored or reviewed drafts of the paper, and approved the final draft.

Campbell R. Sheen conceived and designed the experiments, performed the experiments, authored or reviewed drafts of the paper, and approved the final draft.

James M. Dickson performed the experiments, authored or reviewed drafts of the paper, and approved the final draft.

Catherine L. Day conceived and designed the experiments, authored or reviewed drafts of the paper, and approved the final draft.

Lauren H. Carlton performed the experiments, authored or reviewed drafts of the paper, and approved the final draft.

Prachi Sharma performed the experiments, authored or reviewed drafts of the paper, and approved the final draft.

J. Shaun Lott conceived and designed the experiments, authored or reviewed drafts of the paper, and approved the final draft.

Barbara Koch performed the experiments, authored or reviewed drafts of the paper, and approved the final draft.

Julie Bennett performed the experiments, prepared figures and/or tables, authored or reviewed drafts of the paper, and approved the final draft.

Michael G. Baker conceived and designed the experiments, authored or reviewed drafts of the paper, and approved the final draft.

Stephen R. Ritchie conceived and designed the experiments, authored or reviewed drafts of the paper, and approved the final draft.

Shivani Fox-Lewis performed the experiments, analysed the data, prepared figures and/or tables, authored or reviewed drafts of the paper, and approved the final draft.

Susan C. Morpeth conceived and designed the experiments, authored or reviewed drafts of the paper, and approved the final draft.

Susan L. Taylor conceived and designed the experiments, authored or reviewed drafts of the paper, and approved the final draft.

Sally A. Roberts conceived and designed the experiments, authored or reviewed drafts of the paper, and approved the final draft.

Rachel H. Webb conceived and designed the experiments, authored or reviewed drafts of the paper, and approved the final draft.

Nicole J. Moreland conceived and designed the experiments, analysed the data, authored or reviewed drafts of the paper, and approved the final draft.

The following information was supplied relating to ethical approvals (i.e. approving body and any reference numbers):

Human plasma and sera were obtained from several different sources, all of which were granted ethical approval by the University of Auckland Human Ethics Committee or the Health and Disability Ethics Committee in New Zealand (UOA021200; HDEC 17/STH/233; HDEC 17/NTA/262; HDEC 20NTB76).

The following information was supplied regarding data availability:

The raw data from the 2-step ELISA and the sVNT assays are available as Supplemental Files.

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
