# Peer review of "Collaborative networks enable the rapid establishment of serological assays for SARS-CoV-2 during nationwide lockdown in New Zealand"

_PeerJ, doi:10.7717/peerj.9863_

## Round 0.1 · original submission · Major Revisions

Three reviewers, who are experts in the field, have evaluated your manuscript, and their comments are attached here. While two reviewers have several minor issues to be addressed, the other reviewer raises a major concern regarding the novelty of the study and the methodology (i.e., demonstration of the correlation of the 2 serological tests).

Apart from that, I wonder if the authors could clarify and elaborate on some issues below:

- Did the authors perform within- and between-run ELISA analyses of reference samples to see how reliable and reproducible the result is?

- Fig 1C and 1D, I wonder if adding the levels of each Ig subclass from the control samples will be useful and informative, so readers can see the differences of these values between the case and control groups.

- Fig 1C and 1D, please check if the Y-axis represents AUC, not OD nor titer? If the Y-axis represents AUC, how AUC was calculated?

- How was the ELISA cut-off value (i.e., 0.2) calculated? When the authors perform ELISA at different time points (i.e., on different days), do the authors use the same ELISA cut-off value (i.e., 0.2)? If so, how the authors can be sure that the between-run results are comparable and the cut-off value is still valid each time (giving that ELISA OD of the same serum sample could fluctuate each run)?

- In the method part, the protein expression protocol is worth mentioned briefly since the authors modified the original method of Stadlbauer et al 2020.

If you decide to revise the manuscript, please respond to all the comments point-by-point, before the manuscript is further considered for publication.

Reviewer 1 ·

Basic reporting

no comment

Experimental design

McGregor et al. showed the collaborative work in New Zealand for the rapid establishment of serological assays for SARS-CoV-2. Although the manuscript nicely described the effort to produce the in-house test kit quickly, the work did not provide the novelty to the field. The antigens were produced from the provided vectors, and the assays were performed using ELISA. Another major point is that the authors compared the in-house test to the surrogate viral neutralization test (sVNT). The preprint by Tan et al. analyzed only 13 samples to demonstrate the correlation between sVNT and the standard neutralization test. If the BSL-3 facility is not available, and the authors would like to demonstrate the correlation of serological assays with the viral neutralization, the comparison should be better performed using validated pseudotype virus.

Validity of the findings

no comment

Reviewer 2 ·

Basic reporting

No comment

Experimental design

No comment

Validity of the findings

No comment

Additional comments

This is a well-structured and conducted study that is clearly laid out and written. It is a preliminary evaluation of the testing but adds solid information to the usefulness of these antigens, specific Ab isotypes and platforms for serology to support retrospective case finding, evaluating the tests using human samples.
My only suggestion is that perhaps a mention could be made of the impact of repeated testing (using a new blood sample) on improving issues around sensitivity.
I wonder if the authors could note whether any of their negative sera were from patients with respiratory virus infections known to have been due to an HCoV and if there was anything notable about those results compared to non-HCoV positives.

·

Basic reporting

no comment

Experimental design

no comment

Validity of the findings

no comment

Additional comments

The authors had done a commendable effort in difficult time of COVID-19 pandemic when medical testing were lacking in the situation of many countries locking down. The article validated the in-house ELISA for IgG, IgM, IgA to SAR-CoV-2 RBD and spike1 using 3 sera or plasma from 3 groups; a. COVID-19 PCR positive patients, b. PCR negative COVID-19-like presentation, and c. sera obtained before the pandemic. Showing the best performance, the IgG ELISA was utilized in a 2-step process; screening with anti-RBD followed by anti-spike confirmation. Outstandingly, the detection of a PCR-negative case demonstrated the usefulness and necessity of serology for complimenting the PCR.
The results were also compared with a commercial surrogate neutralization antibody test (sVNT), of which the format is a total anti-RBD detection using blocking assay. As expected, the IgG anti-RBD showed the most correlation with sVNT. This article also revealed the high specificity of sVNT.

Specific items for the authors:
1. “Alert 4” (line 64, 231), in the New Zealand COVID-19 Alert Levels, is needed to be clarified for reader outside New Zealand.
2. Please discuss more of the reason why false-positive results in anti-RBD screening became negative in anti-spike test.
3. The authors did not find good performance of IgA testing. Considering lower level of IgA than IgG, should a higher concentration of anti-IgA peroxidase be used? Please also discuss regarding the finding of a commercial IgA anti-spike being better than IgG. (Kathleen G. Beavis, Scott M., et. al. Evaluation of the EUROIMMUN Anti-SARS-CoV-2 ELISA Assay for detection of IgA and IgG antibodies. Journal of Clinical Virology. 2020;129:104468, ISSN 1386-6532, https://doi.org/10.1016/j.jcv.2020.104468).
4. New reference showing sVNT correlation with conventional VNT was available and should be added (Tan, C.W., Chia, W.N., Qin, X. et al. A SARS-CoV-2 surrogate virus neutralization test based on antibody-mediated blockage of ACE2–spike protein–protein interaction. Nat Biotechnol (2020). https://doi.org/10.1038/s41587-020-0631-z)

---

## Round 0.2 · accepted · Accept

All concerns raised have been properly addressed, and the revised manuscript is suitable for publication in PeerJ.